# Scaling and data collapse from local moments in frustrated disordered quantum spin systems

Itamar Kimchi [1], John P. Sheckelton[2], Tyrel M. McQueen[2,3] & Patrick A. Lee[1]

Recently measurements on various spin–1/2 quantum magnets such as $H_3LiIr_2O_6$, $LiZn_2Mo_3O_8$, $ZnCu_3(OH)_6Cl_2$ and 1T-$TaS_2$—all described by magnetic frustration and quenched disorder but with no other common relation—nevertheless showed apparently universal scaling features at low temperature. In particular the heat capacity $C[H, T]$ in temperature $T$ and magnetic field $H$ exhibits $T/H$ data collapse reminiscent of scaling near a critical point. Here we propose a theory for this scaling collapse based on an emergent random-singlet regime extended to include spin-orbit coupling and antisymmetric Dzyaloshinskii-Moriya (DM) interactions. We derive the scaling $C[H, T]/T \sim H^{-\gamma}F_q[T/H]$ with $F_q[x] = x^q$ at small $x$, with $q \in \{0, 1, 2\}$ an integer exponent whose value depends on spatial symmetries. The agreement with experiments indicates that a fraction of spins form random valence bonds and that these are surrounded by a quantum paramagnetic phase. We also discuss distinct scaling for magnetization with a $q$-dependent subdominant term enforced by Maxwell's relations.

[1] Department of Physics, Massachusetts Institute of Technology, Cambridge, MA 02139, USA. [2] Department of Chemistry, Department of Physics and Astronomy, and the Institute for Quantum Matter, The Johns Hopkins University, Baltimore, MD 21218, USA. [3] Department of Materials Science and Engineering, The Johns Hopkins University, Baltimore, MD 21218, USA. Correspondence and requests for materials should be addressed to I.K. (email: ikimchi@gmail.com)

Heat capacity and its temperature and magnetic field dependence is a powerful tool to provide fundamental thermodynamic information on various solids including correlated electrons in magnetic Mott insulators. It has come to our attention that recent measurements of the heat capacity of certain quantum magnets which are candidates for an exotic state of matter called quantum spin liquid[1] show a power law temperature dependence and a striking one-parameter scaling and data collapse as a function of temperature $T$ and magnetic field $H$. Though the materials all appear quite different the observed scaling functions of $C[H, T]$ as power laws in $T/H$ suggest an unexpected hidden universality.

Power-law specific heat is a familiar consequence of the 'random singlet' phase seen in doped semiconductors such as Si:P, and described theoretically in 1D by Dasgupta, Ma and Fisher[2,3] and in 2D and 3D by Bhatt and Lee[4–6]. In this picture the spins interact with each other via a broad distribution of antiferromagnetic exchange interactions. The spins that are most strongly coupled pair into singlets first, leaving behind spins that are further apart, eventually resulting in a power law distribution of exchanges and a power law tail of density of states. This picture follows from renormalization group analysis in 1D and has been demonstrated numerically in higher dimensions under a variety of conditions. Though the original setting for the $D > 1$ random-singlet phase required a dilute random network of spin–1/2 sites, without a lattice, recently a random-singlet regime has been argued to arise as a general feature of spin–1/2 lattice magnets with quantum paramagnetic ground states and random exchange energies, i.e. in highly-frustrated quantum magnets with quenched disorder[7]. The theory applies when the majority of spin–1/2 sites form a paramagnetic state such as a spin liquid or a valence bond crystal, and demonstrates in two different controlled limits that a small fraction of sites necessarily nucleates (as may be required by the conjectured disordered-Lieb-Schultz-Mattis restrictions[7]) and leads to a random network of spin–1/2 moments at low energies. This small subsystem may be captured by a random-singlet regime in its low temperature renormalization group flow, developing a power-law probability distribution of antiferromagnetic exchange energies $P[J] \sim J^{-\gamma}$. At a given temperature $T$, the spins with exchange $J < T$ behave as free spins giving rise to a heat capacity $C[T] \sim T^{1-\gamma}$ and spin susceptibility $\chi[T] \sim T^{-\gamma}$. In many cases the measurable response from this relatively small emergent subsystem may dominate over the response of the bulk phase. Such an emergent power-law energy distribution, associated with a relatively small portion of the spin–1/2 sites, serves as the a priori starting point for the present work.

The power-law distribution of entropy manifests most dramatically by varying both temperature and an external magnetic field. Consider a given singlet bond with singlet-triplet energy splitting $J$ drawn from the distribution $P[J]$. Applying a magnetic field with Zeeman energy $H$ has no effect on the singlet energy but splits the triplet manifold. At low temperatures $T \ll H$ this bond contributes to heat capacity only when its ground state crosses over from the singlet state to the field-polarized triplet state: the width of this resonance $|J - H| < k_B T$ (Fig. 1) is set by temperature giving a heat capacity that rises linearly in $T$. The distribution of bond energies enters only through the $H$-dependent coefficient: $C \sim T/H^{\gamma}$ at $T \ll H$ where $C \sim T^{1-\gamma}$ at $H = 0$. (We have set the magnetic moment $g\mu_B$ to be unity).

Given this expected scaling it came as a surprise that new experiments of ref.[8] found a different scaling behavior on the spin–1/2 magnet $H_3LiIr_2O_6$, a member of the family of so-called Kitaev honeycomb iridates[9,10]. Both magnetic frustration and disorder are likely ingredients in this compound: exchanges between $Ir^{4+}$ effective spin–1/2 moments may vary randomly

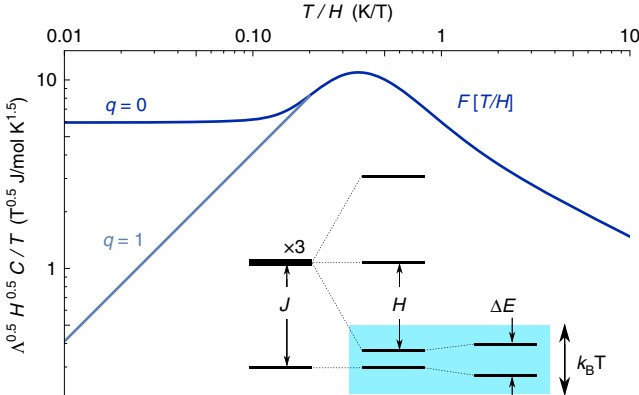

**Fig. 1** Heat capacity scaling function $F_0[T/H]$ and its $q = 1, 2$ modification by level repulsion. The heat capacity $C[H, T]$ for $q = 0$ random singlets (shown with integrated energy distribution $\int P[E] = (E/\Lambda)^{1-\gamma}$ at $\gamma = 0.5$, blue) exhibits scaling collapse in $T/H$ with $T$-linear form at $T \ll H$. This is easily understood (bottom inset). A spin–1/2 pair with singlet-triplet splitting $J$ acquires a resonance in magnetic fields $H \approx J$, contributing to $C[H, T]$ when $|J - H| < k_B T$. The scaling is modified when spin-orbit coupling and lattice symmetries combine into DM interactions with singlet-triplet mixing: the resulting level repulsion changes the resonance condition $\Delta E < k_B T$ to produce $T$-scaling with higher powers, as in the $q = 1$ line shown (gray)

depending on the positions of mobile hydrogen ions, and the frustration expected from iridium's strong spin-orbit coupling is evidently manifest in the lack of any ordering transition down to at least 0.05 K, less than a percent of the 100 K exchange energy scale. The bulk of the sites form a quantum paramagnetic phase, such as a spin liquid. The remaining fraction of sites was found to contribute a power-law heat capacity $C \sim T^{1/2}$ with an appropriate small coefficient as expected from a random-singlet regime; but when a magnetic field was applied, instead of $T$-linear heat capacity, clear quadratic scaling $C \sim T^2/H^{3/2}$ was observed for $T \ll H$. Furthermore, as shown in the inset of Fig. 4a in ref.[8], the data are found to collapse to a single scaling curve of the form $C[H, T]/T \sim H^{-\gamma} F[T/H]$ where $\gamma = 0.5$.

The work of ref.[8] reminds us of a system we had studied earlier, $LiZn_2Mo_3O_8$. This is a layered magnet where 2/3 of the spin disappears into singlets, leaving behind 1/3 of the spins which behave almost as free spins[11–13]. The mechanism for this behavior is not well understood, since various theoretical proposals[14–16] all require some form of short range tripling of the unit cell size, which has been searched for and not found[13]. However on symmetry grounds we again expect the prevalent non-magnetic Li/Zn site mixing disorder[11] to generate bond randomness, whose competition with singlets requires[7] low energy spin excitations to appear. Here we focus our attention on the fate of these remaining spins at low temperatures. It turns out that independent of ref.[8], we had recognized that our previously unpublished heat capacity data (Fig. 2) also show clear data collapse with $T^2$ scaling (Fig. 3). Similar data collapse, though with a smaller accessible $T/H$ window, is seen in previous heat capacity measurements from ref.[17] for synthetic herbertsmithite $ZnCu_3(OH)_6Cl_2$ (Fig. 4). These materials, and related ones which we will discuss further below—1T-TaS$_2$, YbMgGaO$_4$, YbZnGaO$_4$ and Ba$_2$YMoO$_6$—differ in their spin–1/2 lattices, magnetic Hamiltonians, spatial symmetries, and sources of randomness; but their lack of magnetic order down to the lowest temperatures measured, the presence of some randomness, and the observed power-law scaling laws consistent with a contribution from a fraction of sites, taken together call for a unified theoretical framework.

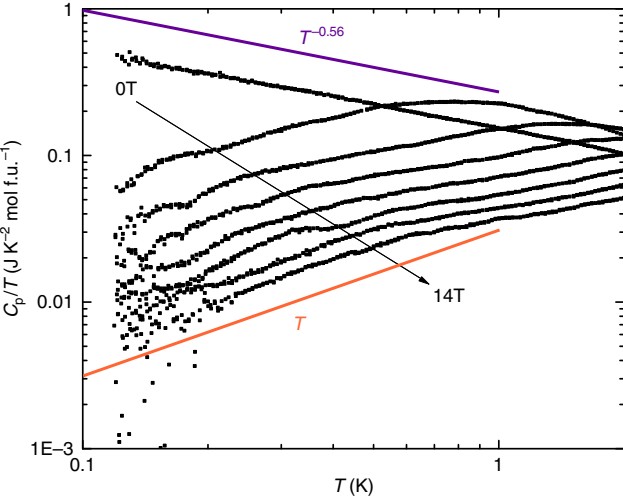

**Fig. 2** Power law heat capacity in LiZn$_2$Mo$_3$O$_8$ under various magnetic fields. At zero field the heat capacity shows a non-integer power law $C/T \sim T^{-0.56}$ (purple line), changing under various magnetic fields to functional forms with quadratic $C \sim T^2$ behavior (orange line) at low temperature

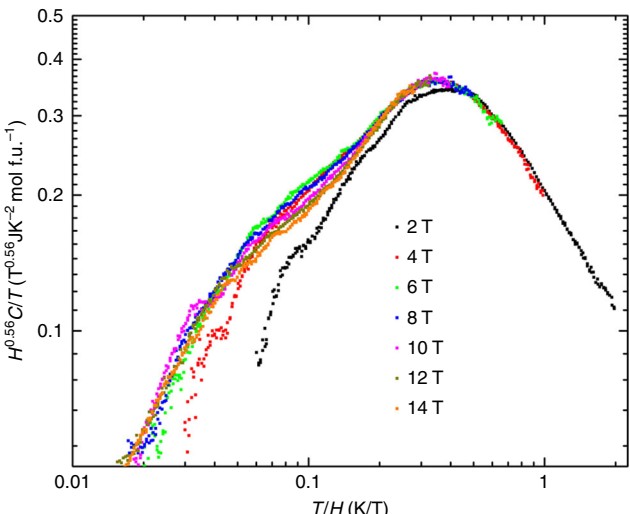

**Fig. 3** Data collapse in LiZn$_2$Mo$_3$O$_8$. Data from Fig. 2 rescaled by the scaling ansatz $H^{0.56}C/T$. The data collapses to a function of the single variable $T/H$, asymptoting as $(T/H)$ for $T \ll H$ and $(T/H)^{-0.56}$ for $T \gg H$, consistent with the $q = 1$ scaling theory Eq. (2) and Fig. 1

In this work we argue that the missing factor of $T/H$ needed to explain the more recent data is captured by an extension of the random-singlet theory that includes spin-orbit coupling (SOC) and its interplay with spatial symmetries, in particular through antisymmetric Dzyaloshinskii-Moriya (DM) spin exchanges. For the sake of continuity we will denote the theory as 'random-singlets', even though the resulting theory describes a configuration of non-degenerate valence bonds that are no longer singlets under spin rotation. We will show how this produces three possibilities for $T$ dependence of heat capacity in a magnetic field, associated with data collapse in $T/H$ and a scaling exponent that can take one of three integer values.

## Results

**Scaling and data collapse.** The derivation of the $T$-linear scaling shown in Fig. 1 assumed that the downshifted triplet state crosses

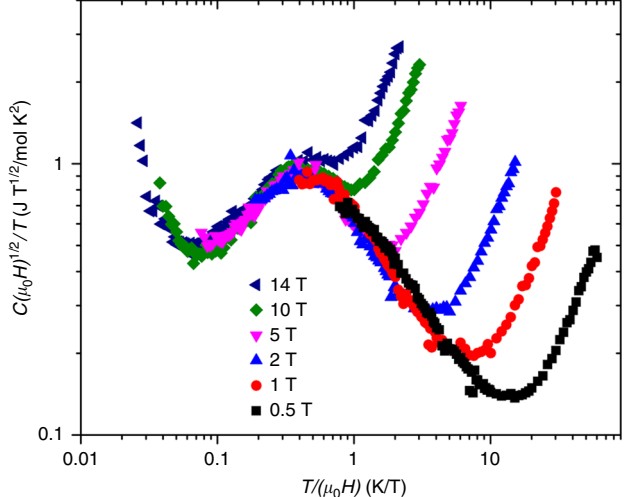

**Fig. 4** Data collapse in synthetic herbertsmithite ZnCu$_3$(OH)$_6$Cl$_2$. Herbertsmithite heat capacity data from ref.[17], here replotted using the scaling ansatz $H^{0.5}C/T$ (per formula unit of ZnCu$_3$(OH)$_6$Cl$_2$) to show data collapse. The phonon contribution has not been subtracted, which accounts for the upturn for $T/H \gtrsim 1$. Away from the upturn at low $T/H \lesssim 0.05$, which is due to the nuclear Schottky contribution, the data collapses with a peak consistent with the scaling function Eq. (2) and Fig. 1

the singlet state freely. This assumption no longer holds in the presence of spin orbit coupling. The triplet will in general be split but in a magnetic field one level will still move down to approach the ground state with a modified $g$ factor. Any off-diagonal matrix element $D$ will mix these states, leading to level repulsion and resulting in an additional, independent, constraint $D < T$ supplementing the diagonal constraint $|J - |H|| < T$. It is important to note spin-orbit coupling is a necessary but not sufficient ingredient for the level repulsion: spin-orbit coupling modifies the magnetic exchanges in two distinct ways depending on spatial symmetries. With an inversion center at the bond midpoint, or certain other combinations of symmetries, the matrix $J_{ij}^{\alpha\beta}$ of the spin exchange $J_{ij}^{\alpha\beta} S_i^\alpha S_j^\beta$ is required to be symmetric. The two-spin singlet state is odd under the inversion symmetry and does not mix with the triplet manifold. However without these symmetries, the spin exchange matrix gains an antisymmetric contribution to $J_{ij}^{\alpha\beta}$ which is conventionally described by the DM term $\mathbf{D} \cdot (\mathbf{S}_i \times \mathbf{S}_j)$. The magnitude of the DM vector $\mathbf{D}$ is linear in the strength of SOC for weak SOC. By breaking inversion symmetry $i \leftrightarrow j$ it mixes the singlet and the triplet and produces the desired level repulsion.

The preceding argument may be considered in more detail. While the zero-field splitting is given simply by $(1/2)\sqrt{J^2 + |D|^2}$, in a magnetic field this splitting becomes (Supplementary Note 1)

$$\Delta E = \frac{1}{2\sqrt{2}} \sqrt{2(|H| - J)^2 + D_1^2 + D_2^2} \qquad (1)$$

where $D_1$, $D_2$ are the components of the DM vector $\mathbf{D}$ that lie perpendicular to the magnetic field $\mathbf{H}$. It is clear that each component of the DM vector $\mathbf{D}$ that enters into the resonance condition Eq. (1) contributes a factor of $T/D$ to the scaling of specific heat at low $T$ (Supplementary Note 3). Furthermore, as long as the ratio of DM interactions to symmetric exchange interactions remains roughly fixed in the RG flow, these factors can be replaced by $T/H$. We show that this is indeed the case within the strong-disorder-RG renormalization step (Supplementary Note 2). The details of the scaling functions can depend on crystal symmetry as well as the dimensionality of the magnetic

lattice (Supplementary Note 4). We thus find three possibilities: the linear scaling $C[T] \sim T/H^{\gamma}$ may stay the same or gain a factor linear or quadratic in $T/H$. This result can be captured by a general scaling function $F_q$ for the density of states as measured by heat capacity,

$$\frac{C[H,T]}{T} \sim \frac{1}{H^{\gamma}} F_q\left[\frac{T}{H}\right]$$

$$F_q[X] \sim \begin{cases} X^q & X \ll 1 \\ X^{-\gamma}\left(1 + \frac{c_0}{X^2}\right) & X \gg 1 \end{cases} \quad (2)$$

The sub-dominant scaling term at large $T/H$ must be added in order to satisfy the Maxwell relation between entropy and magnetization. A related subdominant term must be added to the scaling of magnetization at $T \ll H$ yielding $M/H \sim H^{-\gamma}(1 - m_1(T/H)^{q+2})$ (for $m_1$ and further discussion see Supplementary Note 5). Here the coefficient $c_0$ is given by $c_0 = ((1 + \gamma)\gamma/2)/(C_{H=0}/T\chi_{H=0})$ when the zero-field susceptibility is $\chi_{H=0} \sim T^{-\gamma}$. This scaling form shows the familiar non-universal exponent $0 \lesssim \gamma \lesssim 1$ that characterizes the random-singlet distribution, but in addition a new integer index $q$ appears that takes one of the three values $\{0, 1, 2\}$ for the three cases described above; its choice depends on the interplay of SOC with spatial symmetries as we will discuss below.

Two different values of $q$ are necessary to capture the observed experimental scaling we are aware of so far. We have already discussed the quadratic scaling ($q = 1$) observed for $H_3LiIr_2O_6$ and $LiZn_2Mo_3O_8$. The $q = 1$ scaling in these layered magnets may be understood as arising from a DM vector that is forced to point perpendicular to the lattice plane by an approximate reflection symmetry across the plane that emerges during the course of RG flow: in each strong-disorder RG step the new spin exchanges are generated based on the pattern of valence bonds that have been integrated out at higher energies, and for a 2D lattice any such configuration of valence bonds preserves the mirror reflection across the plane, leading (Supplementary Note 4) to a single-component DM vector and $q = 1$.

In Fig. 4 we show the data collapse for the well studied spin–1/2 kagome lattice material $ZnCu_3(OH)_6Cl_2$. It is known that approximately 15% of the Zn sites which are off the kagome planes are replaced by Cu, forming $S = 1/2$ local moments[18]. Furthermore, there is significant DM coupling. We should mention two caveats. First the nuclear contribution makes the low temperature spin contribution inaccessible and second, the bulk spin gap is estimated to be 0.7 meV, so that the high field data may have some bulk contribution due to the closing of the spin gap. With these reservations, the data collapse shown in Fig. 4 may also be consistent with the $q = 1$ case.

Next we consider the layered material 1T-TaS$_2$, where a charge density wave (CDW) transition at intermediate temperatures leads to a cluster Mott insulator with one spin–1/2 per 13-site unit cell arranged into a triangular lattice, with phenomenology consistent with a spin liquid state[19]. A small fraction of the spins (less than than a few % per cluster) were recently observed to produce a low temperature $T$-linear term in the heat capacity with a coefficient that decreases monotonically with a magnetic field (ref.[20] Fig. 4) roughly as $C \sim H^{-2/3}T$. This is consistent with the argument above for linear scaling ($q = 0$) for 1T-TaS$_2$ due to its high crystal symmetry, preserved even in the CDW state, that forbids DM interactions from being generated for not only nearest neighbor and second neighbor bonds but further neighbor bonds as well. A replot of the $C[H, T]$ two-parameter data (Y. Dagan and I. Silber, personal communication) indeed shows data collapse consistent with Eq. (2) with $q = 0$.

It will be interesting to test the data collapse scaling form for other frustrated spin–1/2 quantum magnets. Other layered compounds such as $YbMgGaO_4$ and $YbZnGaO_4$, both with effective spin–1/2 moments from Yb arranged on a triangular lattice and with intrinsic Mg/Ga and Zn/Ga charge disorder[21,22] show clear $C \sim T^{1-\gamma}$ scaling in zero field measurements[21,23,24]; but the $T \ll H \ll J_0$ scaling limit, given the small lattice magnetic exchange energy $J_0 \sim$ few K, is not yet clear. The case $q = 2$ is expected to arise most naturally in magnets with fully 3D magnetic lattices such as $Ba_2YMoO_6$[25], which we leave for future work.

Finally, we note that the appearance of scaling and data collapse sheds light on two interrelated subsystems of the quantum magnet. The first subsystem is formed by the local moments that contribute to the heat capacity scaling. When the material disorder does not directly change the sites of its spin–1/2 lattice but rather impacts it (weakly or strongly) only through bond randomness in the magnetic exchanges—likely the case for all the materials discussed above except synthetic herbertsmithite —then the local moment subsystem is emergent via an unusual RG flow[7]. The quantum critical scaling functions which can be exhibited by this subsystem are interesting both in their own right and as a signature of a coexisting quantum paramagnet state for the second subsystem, consisting of the remaining spins. This coexisting quantum paramagnetic phase must involve valence bonds which may be frozen, possibly with a relic of valence-bond-solid order, or resonating, as in a quantum spin liquid and associated topological order. The interplay of the quantum paramagnet with the local moments that produce the $T/H$ scaling merits further exploration.

## Data availability

The data generated in this study are available from the authors on reasonable request.

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

## Acknowledgements

IK thanks Adam Nahum and T. Senthil for discussions and instrumental related work, and Martin Mourigal for relevant discussions. TMM and JPS acknowledge Oleg Tchernyshyov for useful discussions. We thank Joel Helton and Young Lee for discussions and for permission to show their data collapse plot Fig. 4. We thank Hidenori Takagi for discussions. We also thank Yoram Dagan and Itai Silber for sharing their data and plots for 1T-$TaS_2$. IK acknowledges support by the Pappalardo fellowship at MIT. The work at IQM was supported by the U.S. Department of Energy, office of Basic Energy Sciences, Division of Materials Sciences and Engineering under Grant No. DEFG02-08ER46544. TMM acknowledges support of the David and Lucile Packard foundation. PAL acknowledges the support of DOE Grant No. DE-FG02-03-ER46076.

## Author contributions

J.P.S. and T.M.M. contributed to the synthesis and measurements of $LiZn_2Mo_3O_8$. I.K., T.M.M., and P.A.L. contributed to the conception and analysis of the theoretical problem and to the writing of the paper.
