## [Peer Review File · Nature Communications]

Reviewers' comments:

Reviewer #1 (Remarks to the Author):

In this work, Kimchi and collaborators argue that spin-orbit coupling and Dzyaloshinskii-Moriya interactions play a fundamental role within the random-singlet regime of disordered quantum magnets. They propose a modified scaling relation for the specific heat, in order to obtain a collapse of experimental results on various materials.

In my opinion, the paper is interesting, however, I have a few remarks that must be addressed, in order to understand the real impact of these results.

1) First of all, the data collapse in their figures (3 and 4) is limited to few decades (in some cases, even one or less!). In this sense the evidence is not really convincing. Since there are samples with different degree of disorder, it would be interesting to understand whether by increasing the disorder a better data collapse is obtained.

2) The crucial point in their approach is given by the fact that q in Eq.(2) can be equal to 1. However, I cannot understand if this is a true outcome of the calculations (from a strong disorder RG procedure) or is a phenomenological fact that is imposed a posteriori. I encourage a more detailed discussion on this aspect, making clear what are the outcomes of the real calculations and what are the more phenomenological parts.

In summary, I think that the paper could contain some interesting issues, but still it is not completely convincing at this stage.

Reviewer #2 (Remarks to the Author):

In this manuscript, the authors calculate the heat capacity in magnetic field of quantum spin liquid states, which have been recently reported in several classes of materials. The authors show that the heat capacity data collapse into a single curve if they are scaled by power laws in T/H . This prediction is based on the model, in which isolated spins that appear as a result of randomness form random singlets. These random singlets are surrounded by a quantum spin liquid phase.

Specifically, the authors show the universal curve in $\text{LiZn}_2\text{Mo}_3\text{O}_8$, which has 2D triangular layers of MoO_3 clusters. In this compound, the authors suggest the importance of Li/Zn site mixing, which induces the randomness in the 2D plane. In perfect kagome compound, $\text{ZnCu}_3(\text{OH})_6\text{Cl}_2$, the authors also show the scaling curve, suggesting the importance of the randomness induced by the replacement of Cu by Zn. The authors also mention the scaling curve in recently reported triangular compound 1T-TaS_2 .

Quantum spin liquids are novel state of matter, which has aroused great interest recently, and the understanding of the specific heat is of primary importance. Therefore the subject in this manuscript is highly topical. The proposed scaling relation provides important progress towards the understanding of the elusive quantum spin liquid states. I therefore recommend the publication in Nature Commun.

I have some minor comments that the authors should address before the publication.

1) To understand this manuscript, reading of Ref. 7 is necessary. More detailed explanation, including a schematic figure, would help the readers of Nature Commun. understand the physical picture of random valence bonds surrounded by a quantum paramagnetic state.

2)The authors assume that the surrounding quantum spin liquid state is fully gapped and does not contribute to the specific heat at low T and H. However, for instance, the presence/absence of gap in herbertsmithite is controversial. Moreover, in many candidate materials of quantum spin liquids, it is not settled. The authors should clearly mention this assumption.

Reviewer #3 (Remarks to the Author):

This paper presents a simple strong-randomness description of the local moments in disordered frustrated magnets with spin-orbit coupling. This simple theory is then compared with data on various quantum magnets, and appears to be in good agreement. I do not see any obvious objections to the theoretical approach, and the data collapse is reasonably convincing, so I think the paper is suitable for publication. However I found the paper somewhat confusingly organized (especially for those who have not worked on the specific materials): the introduction is overly long and detailed. A more concise introduction followed by a description of the theory and then an explanation of how it applies to all the specific cases might make the paper more accessible.

Response to Referees

We sincerely thank all three referees for their time and effort in their review of our manuscript. Based on the various comments and suggestions we undertook an extensive revision of the manuscript. Among various revisions, most significantly we performed and incorporated an additional theoretical calculation suggested by Referee 1. The revised manuscript is attached.

Below see the point-by-point responses to the various comments. Revisions to the manuscript are highlighted as part of the point-by-point response.

(I) Response to comments raised by Reviewer 1:

(1) “First of all, the data collapse in their figures (3 and 4) is limited to few decades (in some cases, even one or less!). In this sense the evidence is not really convincing. Since there are samples with different degree of disorder, it would be interesting to understand whether by increasing the disorder a better data collapse is obtained.”

There are two comments here. The first is a concern about the experimental evidence for scaling, namely whether it is limited to “one or less decades” and thus whether it is convincing.

The data collapse we show in Figure 3 for $\text{LiZn}_2\text{Mo}_3\text{O}_8$ covers T/H across the range $[0.015, 1]$, i.e. almost two decades; the data collapse we show in Figure 4 for Herbertsmithite covers the range $[0.04, 4]$, i.e. two decades. The third material we discuss extensively is $\text{H}_3\text{LiIr}_2\text{O}_6$. This material shows arguably the best data collapse of the three cases, across more than two decades in T/H , as can be shown in the figure below, reproduced from the experimental paper:

(Reproduced from Figure 4(a) of Kitagawa et al, *Nature* volume 554, pages 341–345 (15 February 2018), doi:10.1038/nature25482.)

The data collapse shown in the inset looks quite good. Due to the energy scales of this system, to reach $T/B = 0.01$ it was sufficient to apply fields only up to 8 tesla, thereby avoiding complications to the low- T data collapse from nuclear contributions which become visible only roughly above 10 tesla. As similarly discussed in Fig. 4 of our paper, the upturn at large T/B is due to phonon contributions, which have not been subtracted.

These three independent instances of data collapse, from three different compounds, onto the same scaling function, across approximately two decades in each case, should serve in our opinion as sufficiently convincing evidence that real data collapse is occurring.

There is also a second question raised by the Reviewer here: “it would be interesting to understand whether by increasing the disorder a better data collapse is obtained.” We agree this is an interesting theoretical question but it is far beyond the scope of this work. Actually one

controlled theoretical limit discussed in Ref 7 implies that in certain cases the behavior with disorder is non-monotonic, i.e. decreasing the disorder could actually give a wider regime of expected data collapse. This is thus already known to be a highly non-trivial question whose answer is beyond the scope of any single manuscript.

(2) “The crucial point in their approach is given by the fact that q in Eq.(2) can be equal to 1. However, I cannot understand if this is a true outcome of the calculations (from a strong disorder RG procedure) or is a phenomenological fact that is imposed a posteriori. I encourage a more detailed discussion on this aspect, making clear what are the outcomes of the real calculations and what are the more phenomenological parts.”

This is a fair criticism of a shortcoming in our originally submitted manuscript. Indeed it is crucial that q can be equal to 1, and our previous computations indicated this possibility but without rigorously deriving it within a strong disorder RG procedure.

We address this concern by an extensive addition that can be seen in the revised manuscript. As part of this revision we performed a theoretically controlled SDRG computation that yields a fixed point distribution with $q=1$. This addition should make our conclusions more precise and rigorously supported.

This new result can be found in Appendix 2 of the revised manuscript. In particular the new computation, using the setting appropriate for a recursion step deep within an SDRG flow involving Heisenberg and DM interactions on every bond, gives the same DM-independent result for the renormalization of Heisenberg interactions (equation 7), and modifies the renormalization of DM interactions (equation 8) while preserving the separation of scales that is used to both control this SDRG computation and the derivation of scaling shown in the manuscript. The conclusion that $q=1$ is a possible outcome of SDRG remains the same; this new computation thus provides more precise support for the theoretical viewpoint advocated in the paper, which remain unchanged.

The precise meaning of the new theoretical computation is explained by the following discussion, copied verbatim for ease of the Reviewer from the new Appendix 2 in the revised manuscript. The strong disorder RG (SDRG) step entails integrating out a pair of spins and considering the renormalization of interaction for every other remaining pair. We perform it analytically, within a controlled hierarchy of parameters, to establish a recursion condition that must be satisfied by any fixed-point distribution. The recursion condition ensures (1) that the Heisenberg SDRG is not modified by the presence of DM interactions, and (2) that the flow of DM interactions preserves the separation of scales discussed in the main text and used in the derivation of the specific heat scaling.

In one-dimensional systems SDRG is possible to perform analytically in certain cases. 1D systems are special for a variety of reasons, most importantly here since integrating out two neighboring spins in a spin chain results in a system that is still exactly described as a spin chain. Numerical implementations of SDRG in higher dimensional systems, using various approximations, generally find that the SDRG assumptions become uncontrolled and the fate of the ultimate fixed point remains controversial.

Here we avoid such unresolved questions about the fixed point of $d > 1$ SDRG by restricting ourselves to a particular narrow question: how does the presence of DM interactions change the SDRG? We are able to answer this question rigorously by performing a single SDRG step analytically and noting two observations about the resulting recursion relation. First, we find that weak DM interaction does not enter the renormalization of the symmetric (e.g. Heisenberg) exchanges. The SDRG for Heisenberg exchange therefore proceeds identically as in the case without DM interaction. Second, we find that the renormalization of DM interactions preserves a parametric separation of scales between the DM and symmetric exchanges on each bond. The relative scale of the DM interactions therefore does not change and in particular also the control parameter for the first result is preserved under RG. In this precise sense, the DM interactions are merely spectators to the standard SDRG, which proceeds as usual.

(II) Response to comments raised by Reviewer 2:

1) “To understand this manuscript, reading of Ref. 7 is necessary. More detailed explanation, including a schematic figure, would help the readers of Nature Commun. understand the physical picture of random valence bonds surrounded by a quantum paramagnetic state.”

This is a reasonable concern regarding the present manuscript: its results do depend on the conclusions reached in Ref. 7. Knowledge of the full intricacies described in Ref. 7 is unnecessary, but the starting assumption of emergent power-laws in a lattice model needs to be either taken on faith by the casual reader or derived by going through Ref. 7 in detail. We appreciate the Reviewer’s suggestion to add more discussion of the information needed for the casual reader from Ref. 7, including adding a schematic figure. To this end we clarified the relevant discussion in the introduction, and also added a discussion within the Supplementary Material --- as well as a new Figure 5 with a caption that discusses the main ideas from Ref. 7 that are needed to understand the current manuscript. With these changes, we believe the revised version of the manuscript should be sufficiently self-contained.

2) “The authors assume that the surrounding quantum spin liquid state is fully gapped and does not contribute to the specific heat at low T and H. However, for instance, the presence/absence of

gap in herbertsmithite is controversial. Moreover, in many candidate materials of quantum spin liquids, it is not settled. The authors should clearly mention this assumption. “

The Reviewer is correct that we do not compute any contributions from other sources such as possible surrounding spin liquids. In the revised manuscript we strove to make that even clearer, i.e. make it as clear as possible that our derivation of specific heat scaling involves only the contributions to specific heat from the power-law distribution of emergent valence bonds.

We indeed cannot make any statements about any other possible contributions to specific heat, such as from potential gapless spin liquids. The point is that any other contributions should mess up the data collapse, and show differences between the different materials; whatever data collapse, and agreement between the various materials and the theoretical scaling function, is obtained here, it serves as its own evidence that the heat capacity can be sufficiently well modeled by the power law valence bond distribution without making explicit assumptions about any other contributions. This means we may not be able to shed light on the controversy on e.g. whether Herbertsmithite is gapped or gapless, but also that the agreement with our theory in this case means that our conclusions are evidently sufficiently independent of that controversy.

(III) Response to comments raised by Reviewer 3:

“This paper presents a simple strong-randomness description of the local moments in disordered frustrated magnets with spin-orbit coupling. This simple theory is then compared with data on various quantum magnets, and appears to be in good agreement. I do not see any obvious objections to the theoretical approach, and the data collapse is reasonably convincing, so I think the paper is suitable for publication. However I found the paper somewhat confusingly organized (especially for those who have not worked on the specific materials): the introduction is overly long and detailed. A more concise introduction followed by a description of the theory and then an explanation of how it applies to all the specific cases might make the paper more accessible.”

We are glad that Reviewer 3 agrees that the theoretical approach is sound and the data collapse is reasonably convincing. We appreciate the comment about the introduction being overly long and detailed. We have tried to clarify the conclusions in the paper and make the caption of Fig 1 sufficiently clear so that readers who are interested in a concise description of the theory and its application should be able to get a sense of our conclusions by just glancing at the figures. We agree the introduction may be too long for some readers though its length and the context it gives should be helpful for other classes of readers. Hopefully the revised version of the manuscript can better satisfy the various potential (broad) audiences of the paper.

REVIEWERS' COMMENTS:

Reviewer #1 (Remarks to the Author):

I think that the authors gave reasonable answers to my questions and tried to improve the quality of the manuscript. Therefore, the paper can be published.